# The what, why and when of adapting interventions for new contexts: A qualitative study of researchers, funders, journal editors and practitioners' understandings

Lauren Copeland[1]*, Hannah Littlecott[1], Danielle Couturiaux[1], Pat Hoddinott[2], Jeremy Segrott[3], Simon Murphy[1], Graham Moore[1], Rhiannon Evans[1]

1 Centre for Development, Evaluation, Complexity and Implementation in Public Health Improvement (DECIPHer), School of Social Sciences, Cardiff University, Cardiff, Wales, United Kingdom, 2 Primary Care, Stirling University, Stirling, Scotland, United Kingdom, 3 Centre for Development, Evaluation, Complexity and Implementation in Public Health Improvement (DECIPHer), Centre for Trials Research, Cardiff University, Cardiff, Wales, United Kingdom

* copelandlc@cardiff.ac.uk

**Data Availability Statement:** All relevant data are within the paper and its Supporting information files.

## Abstract

### Background

The adaptation of interventions for new contexts is a rapidly developing research area. To date there is no consensus-based guidance to support decision-making and recommend adaptation processes. The ADAPT study is developing such guidance. This aim of the qualitative component of the study was to explore stakeholders' understandings of adaptation, as to date there has limited consideration of how different concepts and meanings shape decision-making and practice.

### Methods

A case study research design was used. Participants/cases were purposefully sampled based on study outcome, study design, expertise, context and country. Semi-structured interviews were conducted with a sample of researchers (n = 23); representatives from research funding panels (n = 6); journal editors (n = 5) and practitioners (n = 3). Data were analysed using the Framework approach. Overarching themes were discussed with the ADAPT study team, with further iterative refinement of subthemes.

### Results

The results generated four central themes. Four themes related to stakeholders' understanding: 1) definitions of adaptation and related concepts; 2) rationales for undertaking adaptation; 3) the appropriate timing for adaptation; and 4) ensuring fidelity when implementing adapted interventions.

**Funding:** This work was supported by UK MRC-NIHR methods panel grant number [MR/R013357/1]. GM, RE, HL PH, SM, and JS received the UK MRC-NIHR grant. The funders had no role in study design, data collection and analysis, decision to publish, or preparation of the manuscript.

**Competing interests:** The authors have declared that no competing interests exist.

## Conclusion

The findings highlight the lack of clarity around key concepts and uncertainty about central decision-making processes, notably why interventions should be adapted, when and to what extent. This has informed the ADAPT study's guidance, shaping the scope and nature of recommendations to be included and surfacing key uncertainties that require future consideration.

## 1.0 Background

The adaptation of interventions for new contexts is a rapidly developing field of research [1–7]. It is largely a response to the fact that using an existing evidence-informed intervention may have efficiency gains over de novo intervention development, effectiveness is often contingent on the context into which it is evaluated [8–10]. There has been a number of case examples of intervention adaptation that provide a justification for its undertaking [2, 11]. While synthesising evidence on the effects of adapted interventions can be challenging due to the limited transparency of processes or reporting [12] adaptations are undertaken and reported, modified interventions tend to be on average more effective than those used without any adaptation [13]. However, there are cases where adaptation can undermine an intervention's impacts. For example, a Swedish adaptation of the Strengthening Families Programme (SFP), which originated from the USA, included changes such as doubling the number of children attending intervention sessions were delivered [14]. While reducing costs and making implementation easier, adaptations undertaken likely altered the dynamics of the intervention in ways which reduced its effectiveness [15].

Frameworks to support the conduct of intervention adaptation have proliferated [1–3, 16]. However, research on adaptation still lacks coherence in terms of commonly used and understood concepts and methods. A recent study [7] of Australian stakeholders' perceptions of relevant frameworks, indicated a need for clarity around the parameters and drivers of adaptation, particularly in terms of what constitutes adaptation, how it might be rationalised, and when it should be conducted. There is a further need to systematically explore the implications of proposed adaptations for intervention delivery and effectiveness in a new context [3, 17, 18]. In particular, stakeholders are likely having to balance the aim of achieving intervention-context fit, while maintaining consistency with the intervention's intended functioning. This is often complicated by the limited reporting of adaptations and their justifications, as it is often difficult to judge whether failures to replicate effects are due to poor adaptations, differences in methodologies used to assess effect, or that targeted mechanisms were not appropriate to the new context [19].

In other areas of intervention development and evaluation, overarching consensus-based guidance has been developed to support research, with the aim of systematising practice and improving methodological robustness. Such guidance has focused on intervention description and development [20], process evaluation [21], effectiveness evaluation [22] and natural experiments [23]. These guidance documents aim to influence the behaviours of stakeholders within complex systems of evidence use and production, including funders, journal editors, researchers who produce research and policy and practice stakeholders who implement its findings in practice. Currently, there is no such guidance for intervention adaptation. To develop such guidance, it is imperative to understand the perspectives of the stakeholders among whom behaviour change is intended so that it has potency in modifying practice [7]. The present

paper explores stakeholders' understandings of intervention adaptation, to inform the development of consensus-based guidance.

## 1.1 The ADAPT study

The ADAPT study (2018–2020) was funded by the UK MRC-NIHR methods panel to develop adaptation guidance [24]. It aims to support researchers, policy-makers, practitioners, funders and journal editors in the funding, conduct and reporting of research on adaptation. The study comprised three work packages: 1) A systematic review of existing adaptation guidance and scoping review of case examples of intervention adaptation [2, 25]; 2) A qualitive study using semi-structured interviews to explore the understandings, perspectives and experiences of researchers, funders, journal editors, and policy and practice stakeholders; and 3) A Delphi expertise consensus exercise to scope the clarity of the definitions and constructs used in the guidance, explore and capture key debates, identify agreement on important adaptation processes, and ascertain areas where there is limited consensus [26].

The work-packages were inter-related. The systematic review [2], scoping review [25] and qualitative study data analysis provided items and issues that required consideration in the Delphi consensus exercise [26]. In particular, the qualitive data analysis informed the concepts that were identified as central to the guidance and definitions that required consultation, both in terms of clarity and usefulness. The qualitative analysis further identified key debates and areas of uncertainty for exploration, reflecting some of the findings of recent reviews [1, 2, 16]. They include a lack of shared understanding on how to define adaptation, uncertainty as to when adaptation is warranted, and a lack of clarity on what aspects of an intervention can be adapted while retaining fidelity.

This paper reports the findings from the qualitative study that underpinned the Delphi exercise and subsequent guidance, providing a more extensive overview of the complexity, nuance and divergences in stakeholders' understandings. It explores the following themes: 1) definitions of adaptation and related concepts 2) the rationales for undertaking adaptation; 3) justifications for the timing of adaptation, including consideration of whether adaptation may be planned or responsive; and 4) the relationship between adaptation and fidelity. These themes provide invaluable insight into why and how stakeholders undertake various adaptation processes in practice.

## 2.0 Methods

### 2.1 Research design

A case study research design was used in the first instance [27–29]. A case of adaptation was defined as a population health intervention that had previously been subjected to adaptation or was currently being adapted. Funders and journal editors were not linked to specific cases but contributed to understanding of the wider evaluation context. As the study unfolded it became increasingly challenging to recruit multiple and varied participants per case. To redress this shortcoming, more emphasis was placed on exploring diverse perspectives across different stakeholders linked to different adapted interventions. The following methods are reported in compliance with all 32 items of the Consolidated criteria for reporting qualitative research (COREQ) [30].

### 2.2 Sample frame

**2.2.1 Adaptation stakeholders (researchers, policy makers, Patient and Public Involvement (PPI) and practitioners).** In total 312 studies reporting intervention adaptations were

identified through two mechanisms: search retrievals that informed both the ADAPT study systematic review of adaptation guidance [2] and scoping review of case examples [25]; and expert recommendation. Trial registries (National Institute for Health Research Applied Programmes Grant; ISRCTN Registry; and Research Register) were searched to identify in-progress studies so as to include the most recent examples of adaptation. A matrix classified adaptations according to: the socio-ecological domain where the theory of change primarily operated (mico, meso or macro); intervention outcome (e.g. obesity, mental health) the contexts between which the intervention was transferred (e.g. country to country, population to population); study design (e.g. effectiveness or feasibility); and evaluation outcome (i.e., favourable or unfavourable). The intended sample size was 15 adaptation cases, although this was responsive to the emerging data. For each adaptation case, the stakeholders involved were mapped. These were researchers, policy makers, patient and public (PPI) representatives and practitioners.

**2.2.2 Research funders.** Research funders were identified from international funding boards suggested by the ADAPT study team and by checking funding sources of adaptations included in the matrix. Funders were classified according to their geographical remit (e.g. UK, USA, Australia, Germany) and area of specialism where possible (e.g. global health, health services). Thirteen funding panels were identified.

**2.2.3 Journal editors and peer reviewers.** Relevant international journals publishing adaptation studies were identified from the matrix. Journals were classified according to country of publication (e.g. USA, Europe and Australia) and area of specialism (e.g. AIDS, implementation science). In total 48 journals were identified.

## 2.3 Recruitment and sampling

**2.3.1 Researchers, policy makers, PPI and practitioners.** Adaptation cases were purposefully sampled from the matrix to achieve variation in the socio-ecological domain of the theory of change, contexts, study design, and outcome. For the stakeholders linked to each of these adaptation cases, snowball sampling was used. First the corresponding research author was invited to participate in an interview. They were asked to recommend policy-makers, practitioners and PPI representatives linked to the study. Initially research authors linked to thirty adaptation cases were approached via email to introduce the study and outline the requirements of participation, with the expectation that a number would not respond. They were emailed up to three times. If there was no response or the individual declined, the intervention was replaced in the sample frame until sufficient participants had been recruited. Due to this rolling recruitment process a total of 73 researchers and practitioners were contacted. The snowball sampling was largely ineffectual in identifying additional stakeholders, and so alternative recruitment routes were tested. This included contacts suggested by the ADAPT study team, advertising through the Involving People charity, which aids public and patient involvement in research, and Twitter promotion targeting the European Society for Prevention Research and the Society for Prevention Research.

**2.3.2 Research funders.** Research funders were purposively sampled from the sample frame to achieve maximum variation in geography and specialism. Sampled funders were contacted. The contact details of the chair of the funding panel or other relevant committee member were requested to invite them to interview. The suggested individual was approached via email, which introduced the study and outlined the requirements of participation. Individuals were emailed up to three times to invite them to take part. If they declined, they were asked if they could suggest another potential participant. In total 20 funders were invited to participate.

**2.3.3 Journal editors and peer reviewers.** Editors and peer reviewers from the journals included in the sample frame were purposively sampled to achieve a range in country of publication and specialism. The relevant editor in chief or specialist editor was contacted via email to invite them to participate or suggest an appropriate editor or reviewer. They were emailed up to three times. In total 16 journals were invited to participate.

## 2.4 Sample characteristics

In total 23 researchers who were involved in the adaptation of 23 interventions took part in the study (Table 1). Only one practitioner participated who was linked to one of the interventions. Two practitioners were recruited via expert recommendation that were not linked to the sampled cases. Six representatives from funding panels participated, based in the USA (n = 1), UK (n = 3), Germany (n = 1), or with an international remit (n = 1). The five participating journal editors represented global health (n = 2) or public health (n = 3). Their primary publishing location was USA (n = 2), Canada (n = 1), countries across Europe (n = 1) and Australia (n = 1). Of those that did not take part, invitees stated that the subject matter was not relevant to them (n = 6), their workload was too high, (n = 2) or they did not respond (n = 64). The study did not succeed in recruiting PPI representatives or policymakers.

Participants were sampled until 'saturation' was reached. This was interpreted as achieving "information power" [31] and depth and representation across i) the groups of stakeholders that we were able to recruit ii) the socio-ecological domain where the theory of change for the adapted intervention primarily operated (mico, meso or macro), iii) the contexts between which the intervention was transferred (e.g. country to country or population to population within a country), iv) study design (e.g. effectiveness or feasibility) and v) outcomes (i.e., favourable or unfavourable) [32]. Saturation was not reached for PPI and policymakers as we were unable to recruit them.

## 2.5 Data collection

Two members of the research team conducted semi-structured interviews (LC/HL) between April and September 2019, informed by topic guides tailored to each set of stakeholders. Guides addressed the ADAPT study's research questions, in addition to the findings and evidence gaps identified by the systematic and scoping reviews [2, 25]. These gaps were adaptation within macro level interventions, the function of context, fidelity to the intervention functions, nuances around stakeholder involvement and re-evaluation. Topics explored: the definition and meaning of adaptation and related concepts; decision-making and experiences in relation to adaptation, re-evaluation, publication, and funding; and views on adaptation guidance development (see S1 Appendix). Specific questions and prompts were flexibly applied. Interviews were conducted via telephone or Skype and field notes taken. The duration ranged from 40 to 75 minutes. Interviews were audio-recorded and transcribed verbatim by a professional transcription company. Transcripts were reviewed for accuracy and anonymised. Participants were offered the option of reading their transcript for their comment. All interviews with participants were conducted prior to Delphi participation. All participants were also asked if they would like to participate in the Delphi study with some going onto accept this invitation.

## 2.6 Data analysis

Data were analysed using the Framework approach [33], and was undertaken by four members of the research team (LC; DC; HL; RE). The Framework approach was chosen as the matrix allowed us to compare data across the cases and between the three stakeholder groups [33].

**Table 1. Recruited sample characteristics.**

| Stage of Study | Participant (researcher/ policymaker/practitioner) | Type of Intervention (macro/meso/micro) | Research Design | Intervention Outcome Measurement | Contextual Transfer (country to country/population to population/ setting to setting) | Evaluation Outcome |
|---|---|---|---|---|---|---|
| **Adaptation cases with 2 participant** | | | | | | |
| Completed | Researcher and Practitioner | Meso | Feasibility | Diet and exercise | Policy to different settings | Infeasible |
| **Adaptation cases with 1 participant** | | | | | | |
| Completed | Researcher | Meso | RCT | Addictions | Country to country | Effective |
| Completed | Researcher | Macro | Feasibility | Reproductive and child health | Country to country | Feasible |
| Completed | Researcher | Macro | Feasibility | Road traffic injury | Country to country | Feasible |
| Completed | Researcher | Meso | Feasibility | Sexual health | Population to population | Feasible |
| Completed | Researcher | Meso | Feasibility | Sexual health | Population to population | Effective |
| Completed | Researcher | Meso | Feasibility | Hearing | Setting to setting | Feasible |
| Completed | Researcher | Micro | RCT | Parenting | Country to country | Effective |
| Completed | Researcher | Micro | RCT | Weight Loss | Population to population | Effective |
| Completed | Researcher | Micro | Feasibility | Diabetes prevention and management | Population to population | Feasible |
| Completed | Researcher | Micro | Feasibility | Smoking: cessation | Population to population | Feasible |
| Completed | Researcher | Micro | Feasibility | Mental health | Country to country | Feasible |
| Completed | Researcher | Micro | Feasibility | Childhood obesity | Setting to setting | Feasible |
| Completed | Researcher | Micro | Feasibility | Exercise | Population to population | Infeasible |
| Completed | Practitioner | Micro | Feasibility and RCT | Addictions | Setting to setting | Mixed |
| Completed | Practitioner | Micro | Feasibility and RCT | Addictions | Setting to setting | Mixed |
| In progress | Researcher | Meso | RCT | Lung health | Country to country | N/A |
| In progress | Researcher | Meso | RCT | Cancer | Population to population | N/A |
| In progress | Researcher | Meso | Feasibility | Weight loss | Country to country | N/A |
| In progress | Researcher | Micro | RCT | Diabetes prevention and management | Population to population | N/A |
| In progress | Researcher | Micro | RCT | Diabetes prevention and management | Population to population | N/A |
| In progress | Researcher | Micro | RCT | Diabetes prevention and management | Population to population | N/A |
| In progress | Researcher | Micro | Feasibility | Weight loss | Country to country | N/A |
| In progress | Researcher | Micro | Feasibility and RCT | Diet and exercise | Country to country | N/A |

Analysis commenced with data familiarisation, reading and listening to the interviews. At this stage each of the three stakeholder data sets (adaptation stakeholders; funders; journal editors) were treated separately. Two interviews from each data set were coded by the whole team to develop a coding frame. The framework included both a priori codes and in vivo codes. The remaining corpus of the data were coded by a single researcher. New codes that were created were agreed with another member of the team, included in the coding framework and applied to previously coded data. The coding frameworks included the adaptation process, definitions of adaptation terminology and the types of adaptations carried out. In order to enhance reliability, 10% of the data was independently checked by a second researcher (RE/DC). Any disagreements between the two researchers were resolved through discussion. This led to refinement of the definition of some of the existing code. No new codes were created as the

team worked closed together to develop the coding frames. NVivo 10 supported data analysis and storage.

Coded data were charted into a framework matrix. This entailed summarising the data within the fields of the matrix, with illustrative quotations being included where relevant. Matrices for each of the three data sets were created. This was to aid understanding of any differences in perspectives between stakeholder groups, as the ADAPT guidance is intended to provide recommendations that are relevant to each. Data within and across the matrixes were compared and contrasted by two members of the research team (LC; RE) as part of the interpretative process of generating themes. Visual maps were created for overarching themes and subthemes. These maps also captured the nuances within the data (including contrasting cases and links to other meta-themes). Overarching themes were discussed with the ADAPT study team, with further iterative refinement of subthemes. After the ADAPT study's Delphi consensus exercise results were known and areas of consensus and disagreement emerged, further analysis was conducted of the qualitative data to bring insight to Delphi findings.

### 2.7 Reflexivity

LC, HL and DC conducted the interviews and the data analysis. LC and HL are female research associate with PhDs. DC is a female research assistant with a MSc. RE is a female senior lecture with a PhD. All are experienced qualitative researchers who have received training in conducting interviews and thematic and framework analysis. All researchers apart from RE and HL did not have a prior relationship before the study. RE and HL had worked previous on studies together. The participants did not know the researchers prior to the study. The participants understood the researchers were conducting the interviews as part of the ADAPT study in order to understand their experiences of conducting adaptation studies. RE and HL have a mythological expertise in adaptation which may have influenced their interview style and analysis of the data based on her extensive prior knowledge of the area. LC and DC were new to adaptation however have both worked on process evaluations looking at context therefore their focus on context may have biased the interview style and analysis. The interviews were guided by topic guides developed by the wider team which will have negated some of the researcher bias. The analysis was double coded which to negate some of the bias of the researchers.

### 2.8 Ethics

Ethical approval was provided by Cardiff University's School of Social Sciences Ethics Committee (Ref: SREC/3165).

### 3.0 Findings

Participants' understanding of adaptation are presented according to the following four themes: 1) definitions of adaptation and related concepts 2) the rationales for undertaking adaptation; 3) justifications for the timing of adaptation, including consideration of whether adaptation can be planned or responsive; and 4) the relationship between adaptation and fidelity. These themes are central to understanding why and how stakeholders undertake adaptation in practice.

### 3.1 Defining adaptation

Interview data indicated ambiguity over how adaptation is understood. Definitions tended to centre on the need to make modifications to an intervention to ensure it is suitable for a new

context. The existing evidence-base, mainly the lack of transferability of the evidence-base across contexts, was occasionally mentioned as being relevant:

> *"So I'm thinking of the implementation world where the generalisability from one setting might not be as relevant once you spread it and so how do you think about what pieces of the intervention are core elements of the. . . . . . the intervention what pieces are adaptable and how to keep the core essence of what was done in one setting but be able to translate that into make it relevant for a new group."*

> *P006 researcher micro-feasible*

For the large part however, participants did not or could not clearly define adaptation. It was often used inter-changeably with other terms, such as *cultural re-grounding*, *contextual sensitising*, *implementation* and *refinement*. A journal editor observed how some constructs were inextricably linked and overlapping, citing adaptation and implementation as a key example:

> *"So yes . . .there is a lot of work that is being done in implementation to fit an idea that has been tested somewhere else to a new context . . .and the terms implementation and adaptation as interchangeable or. . ..I mean there's no implementation without adaptation."*

> *P003 Journal Editor*

There were also difficulties in drawing parameters around research that could be defined and demarcated as adaptation. Some participants recognised this challenge, in particular how adaptation is differentiated from research undertaking de novo intervention development:

> *"When is adaptation, adaptation, and when is it like just developing a new intervention, where's the cut off and how do you assess that?"*

> *P019 researcher meso-in progress*

Equally, others maintained that most interventions involve a degree of adaptation as they derive principles or theories from the existing evidence base of evaluation research. For example, one researcher felt strongly that no interventions are completely novel:

> *"I mean me personally, I'm not. . .I don't think there's anything in public health that is starting from scratch. That's just my personal belief. I don't think I would. . .I don't know what would qualify as starting from scratch. I think it's always good to start with at least some evidence basis for what you're doing.."*

> *P016 researcher meso infeasible*

Meanwhile a funder observed that it was difficult to differentiate between adapted and 'off the shelf' interventions in practice, as both would require some effort to make them contextually relevant:

> *"Yeah. I mean the whole . . . as you know better than I since you're studying this adaptation, adaptation is variably defined and you know what is an adapted guideline versus one that's really adopted off the shelf almost, cos you know everything needs . . . you know everything is*

*contextual when you bring it to the local or national level. It's contextual. You have to make some modifications."*

*P001 Funder*

There was further overlap between the notion of adapting an intervention as it is transferred between contexts and modifications that are undertaken during the scale-up of intervention for routinised implementation following evaluation. Indeed, one researcher defined both of these as instances of adaptation:

*"So then when you try and unpick adaptation, it's a can of worms. There's the one you're adapting, which is taking something that's worked in a country and thinking about how it needs to be adapted from almost the academic trial way of delivery, out into a more serviced, naturalistic situation."*

*P014 researcher micro-in progress*

Other areas of conceptual confusion related to translation. A journal editor discussed the potential of translation to include adaptation, with the former focusing on process of modification while and the latter focusing on the modification of objects or 'things':

*Participant*: *"Well to me that's what we could provide, are general definitions because to me translation very much includes adaptation. . . and so they're not the same thing, but I think there's components within each that inform the other."*

*Interviewer*: *"Yeah, so they're not mutually exclusive?"*

*Participant*: *"No absolutely not and so in a sense maybe it's translation of stuff and adaptation of things and that's the bigger grouping if you know what I mean. . ."*

*P001 Journal Editor*

As a consequence of the lack of clarity around the term adaptation, and how it relates to adjacent constructs, there was suggestion that work is needed to develop some broad, consensual understanding of the terms in use. One researcher felt this would support research in achieving more coherency:

*"I'd like to see some very clear definitions of what the different forms of adaptations were, or a categorisation of adaptations. Which might then move forward in terms of informing the literature, enabling one to find and search. People could then start categorising in an appropriate way."*

*P014 researcher micro-in progress*

Hence while the field of research on intervention adaptation is somewhat emergent, and terms are still in the process of being defined, there is a sense that initial consensus in key areas might make the evidence base more accessible and comprehensible for stakeholders.

## 3.2 Rationales for adaptation

A number of rationales were cited across the stakeholder groups to justify adaptation. The primary reason centred on the need to respond to differences between contexts and cultural

specificities, such as ensuring religious sensitivity or accommodating people with disabilities. Some participants went so far as to suggest that any transfer of interventions across contexts made adaptation inevitable. One journal editor working in global health maintained that adapting, or 'contextualising', interventions to new contexts is so common that it is actually unremarkable, *"it doesn't stand out for me. . ...." P002 Journal Editor*.

Focus on adapting to ensure contextual responsiveness was seen to be driven by two key factors. First, there is the recognition that intervention effects are contextually contingent:

*"Basically it is naïve given what we're increasingly understanding at a systems level about how that context can significantly impact what happens on the ground so to speak, so I think that, the process of ensuring that through adaptation contextual factors are brought in is critical to intervention delivery in other situations and the, if not the literature then the world is full of interventions that are simply being transferred from one place to another not anything like as well as expected."*

*P002 Funder*

Second, the funding climate was seen to encourage the privileging of contextual differences rather than similarities in order to warrant a research study proposing adaptation and re-evaluation. In this sense, some funders acknowledged that applicants are compelled to prove that social equipoise exists, and that the evidence-base from the original evaluation context is not generalisable:

*"So I think it's, the stronger applications are going to say this intervention's worked brilliantly in this context, but we're uncertain, there's equipoise as to whether it will work in another environment and the work that we're proposing seeks to,. . ."*

P002 *Funder*

While acknowledging the importance of being sensitive to context, some participants felt it was potentially problematic. At its most fundamental level, there was uncertainty about how to actually define and operationalise context or adjudicate what it means for contexts to be different or similar:

*"When you say any new context though, as I say, sometimes it's quite difficult to define what is a new context and what is such a similar context, that you may not need to, but doing that scale of research."*

*P013 researcher micro-in progress*

One researcher maintained that such a focus on contextual differences could mean a drift into adaptation hyperactivity where the need of the new context is seen as so specific that the intervention ends up being adapted 'down to a thousand microcosms'. This was seen as a particular issue given the finite resources available for adaptation:

*"Well I think that there is a real question, at least in my mind, when we talk about adapting interventions, it's like how far do you go in terms of adapting, because is it practical to take one intervention and adapt it down to a thousand microcosms, we don't necessarily have the resources for that, so I think you know, some sort of practical guide for clinicians, or public health individuals, who want to take an evidence based intervention and adapt it for a local*

*community. What are five things that they should do? Or ten things that they should do? To make sure that it's going to be successful."*

*P008 researcher micro-in progress*

Equally, there was expressed concerned that adapting interventions in order to accommodate the individual characteristics of one context would make it impossible to assess the wider applicability or generalizability of its evidence base. One researcher commented on this tension between making interventions both tailored and generalizable:

*"I think sometimes we take the adaptation so far out that it's great for that specific population but it's not generalisable to anybody else. So, it's like you've got this evidence-based programme that maybe not generalise to these specific communities, but then you adapt it so much that it's only applicable to that thousand people. So, I think there's a real, there's like a fine line between those two things and I'm not sure we know where that is, does that make sense?"*

*P008 researcher micro-in progress*

An additional rationale for adaptation was the need for interventions to reach systematically excluded population groups and hence reduce health inequalities. A number of researchers recognised that interventions have been developed in a manner that is culturally alienating for a large minority, hence requiring adaptation to ensure sensitivity:

*"I grew up in a poor underserved little village, I think that's a background I kind of take with me, I just want those who need it the most to receive a state-of-the-art science. And so that's my big thing, is this practical like well people actually do this in a way where you can still maintain its effectiveness and won't be. . .can it really be disseminated and cost effective and sustained."*

*P009 researcher micro-effective*

The remaining rationales described by participants tended to focus on potential efficiency gains. For some participants this meant avoiding the continual but arguably unnecessary generation of new ideas, which was seen as simply 'reinventing the wheel':

*"Look, I strongly believe that reinventing wheels (laughs) is meaningless. Coming up with interventions, coming up with new ideas, piloting, it's useless. We have so much examples around the world."*

*P002 researcher macro feasible*

Some participants also considered adaptation to be more efficient and less resource intensive than de novo development, which was seen as particularly important when working in lower resource contexts:

*"And we can't . . . you know it's just not feasible for every country to develop guidelines on type 2 diabetes management. It just makes absolutely no sense."*

*P001 Funder*

However, others raised that adaptation was time-consuming. Some participants maintained that it could take up to a year to iteratively modify an intervention, depending on the level of complexity involved. There was a clear sense that the current funding climate, which often subsides adaptation in the early phases of evaluation, did not permit the required time to fully undertake comprehensive adaptation. This reflects the above opinion by funders that is it less resource and time intensive than de novo development:

*Yes. And so in this particular instance, we were very lucky because the funding was. . .it's very rare I guess, to get funding that is explicitly and exclusively for adapting a campaign. We were done, we handed it over to the (NAME OF FUNDER 1) right? So that kind of funding mechanism is unusual, but it really gave us a chance to do things the right way.*

*P016 11.7.19 SG meso infeasible*

Despite general clarity across stakeholders as to why interventions might be adapted, the process for deciding when adaptation is actually justified was less apparent. Some researchers touched upon their experiences of using frameworks such as RE-AIM [34] or MAP Adaptation [35] to support decision-making, but accounts revealed that these had not been employed in a comprehensive or systematic manner. As such, a number of participants suggested that it was inadequate to simply make stakeholders aware that adaptation may be necessary, and that guidance would be helpful on the processes for making decisions and then undertaking modifications:

*"That's great to say, to recognise researchers, to say that adaptation is important, and you must anticipate it otherwise it will happen anyway but in a way that won't be (?), it won't make the activities effective. Not the fact of saying it but writing it down in black and white so it will be clear for everyone will be great."*

*P027 practitioner meso-feasible*

Within discussion around the need for guidance was also consideration that improved reporting of why interventions are adapted would support stakeholders to make more intentional decisions, and to be more reflective about their justifications, rather than treating adaptation as inevitable or unproblematic.

### 3.3 Timing of adaptation

Participating researchers indicated uncertainty as to the most appropriate time to undertake intervention adaptation. In particular there was a lack of clarity about when in the course of intervention development, evaluation or implementation that modifications are classified as adaptations or refinements. For some participants there was a clear distinction. They indicated an initial period of planned adaptation prior to introducing the intervention to the new context. Refinement is then conducted following implementation in response to emerging information on feasibility and acceptability. One researcher stated that now their intervention was in the new context they had transitioned to a period of *"refining the adapted intervention"* *P015 researcher micro-effective.*

In contrast, other participants did not draw parameters around the period of adaptation as opposed to refinement. A number of researchers discussed the continued need to adapt an intervention as the new context continually evolved, hence conceiving it as an ongoing activity throughout the course of implementation:

*"I think the good thing was just being flexible and being able to continue to adapt. That seemed to be really important for our group. Knowing that it's an ongoing process."*

P007 researcher micro-feasible

*"And now after many years we are adapting it again, but with a different perspective. Now we are adapting it, we work with a group of health professionals, teachers, school principals and the school office representative and we have worked for a couple of years to readapt the programme from a different perspective and these on some school's request."*

P017 researcher meso- effective

Regardless of the nomenclature used, there was consideration of the relative benefits to adapting an intervention prior to introducing it to a new context, compared to modifying it once implementation has commenced. On one hand planned adaptation was seen as particularly productive, especially as it tends to be informed by the extant evidence-base. On the other hand, responsive adaptations were considered most important. This was largely because the intervention only interacts with the new context at this point, and that feasibility and acceptability had started to emerge. Both researchers and practitioners recounted experiences of responsive adaptation once they had seen the level of engagement with participants:

*"So we made the things* [adaptations] *we thought we'd have to make but we decided not to go, and I guess we could have done more, like, our hunch was perhaps we needed to do more in terms of perhaps to really hit acceptability. But in fact, because we were going to have a two-stage pilot, you know, we had an uncontrolled feasibility stage, we'd agreed that we would actually run it with our first bunch of changes, see how it went. And actually get feedback from that to refine it again before we went into a more formal landmark feasibility trial."*

P014 researcher micro-in progress

These perspectives are valuable, as while there can be a propensity to conceive responsive modifications as being part of intervention drift, participants understand them to be intentional and informed by the evidence provided from stakeholders within the new context.

## 3.4 Adaptation and fidelity

Participants explored the nature and extent of adaptations that might be undertaken when transferring an intervention to a new context, considering the point at which extensive adaptation becomes de novo intervention development. Much of this discussion centred on the question of what constitutes the replicable 'intervention'. A number of participants talked metaphorically about "the essence" or "heart", maintaining that this could not be lost if the intervention was to be retained rather than a new approach be developed:

*"So we went through a great big adaptation phase and then it was as if, "Okay, at what point is one throwing baby out with bath water? . . ..What's the essence in that intervention that we don't want to lose?" Because it's a real challenge, actually, because one doesn't know, because one's got this slightly black box, you know, a multi-component intervention."*

P014 researcher micro-in progress

However, unpicking what constitutes this essence is complex. Researchers variously suggested that it could be demarcated by the mechanisms of change, core components, or delivery

strategies. For some working from a more complex systems perspectives, it also extended to include the features of the context in which the intervention is being delivered and their interactions.

In practice, these different aspects of the intervention were often used interchangeably. Commonly, participants felt that the mechanisms of change are the defining feature, and that core intervention components encase them. As one researcher commented, core or 'most important' components serve as the 'heart of the intervention' and are responsible for generating intended outcomes:

> "It really just means changing key components of the curriculum so that it will be more effective with the specific population while maintaining the heart of the intervention or the most important components of the interventions. It differs from creating an intervention; it is both adapting the key components but maintaining the key components as well."

> P011 researcher micro-in progress

For the majority of participants, the intervention components, the delivery strategy and context could be modified, provided that the mechanisms of change were retained. As the 'core components' were often seen as the way in which these mechanisms were operationalised, these were understood to not be malleable. Extending this line of thought, participants also considered the notion of fidelity. In this instance fidelity to the function or mechanisms of the intervention was seen as important, while also ensuring some adherence to the form of core activities to ensure the mechanisms were activated.

In practice, the ability to identify the mechanisms of change, and even core components, was deemed difficult. A participant contrasted the case of complex interventions with clinical drug trials. In the latter instance they felt it is relatively simple to identify the active ingredients, even when using multiple delivery strategies. However, they considered it far more testing to isolate the causal processes within complex, contextually contingent programmes:

> "So the way I think about adapting interventions is really, really important and really, really challenging sometimes to figure out. . . If I were thinking of a medication, you know they list the active and inactive, so whether it's a capsule or tablet, whether it's chewable or it's not, all of those things are maybe adaptable but the chemical components that make it work are not."

> P018 researcher micro-ineffective

There was also some acknowledgment, though only amongst a small number of researchers, that core components might not activate the proposed mechanisms of change in the same manner in the new context. This can make the identification and retention of such components somewhat problematic. In response to this, one researcher encouraged more reflection on how the intervention might interact with wider system dynamics:

> "Well because any intervention is always going to interact with a much wider, a much wider system that delivers within a particular cultural context and you need to step back and think about whether or not the intervention may have different effects, maybe equally acceptable, maybe as easy to implement mainly to different outcomes or adverse consequences and an anticipated one, etcetera."

> P013 researcher micro-in progress

Another suggested that core components may even need adding to the intervention to ensure that the mechanisms could be fully activated in the new context:

*"So in my case I spent quite a bit of my career adapting evidence based interventions that are principle based, and so I am very clear that when I adapt something not everything is up for grabs. But I am keeping the core principles of the intervention whilst modifying the topography for culture and context. And maybe adding in some components where those that need it, given the culture and the context."*

*P015 researcher micro-effective*

There were further pragmatic challenges related to the need to retain mechanisms of change through implementation of core components, namely that they may not actually be deliverable in the new context. Hence stakeholders may have to prioritise the components they believe to have the most leverage, balancing this with the perceived feasibility of implementing components:

*"If they've adapted the intervention and you are then delivering it, there are some components that are more likely to be delivered and other components are less likely to be delivered. We just like to know that because that helps us to maybe make the intervention more lean and easy to train people on, because if you've got lots and lots of different components and lots of different things, more is not always better. . . If there are lots of redundant components, even if they're evidence based we cannot include them in the final version. If nobody's delivering them then what's the point."*

*P026 researcher macro-in progress*

Through consideration of the nature and extent of adaptation, participants noted the paucity of frameworks to support decision-making around fidelity issues. A number of researchers indicated that decisions were currently made based on an assessment of the differences in contexts, populations and outcomes that the original and adapted intervention aimed to address:

*"I guess it's all a matter of gradation. I think if you're going to a very similar target group and say a similar population, let's say you've developed an intervention that's suitable, deemed to be suitable for men of a particular age with particular interests and it maybe that if you're looking across quite similar country context that there wouldn't be, need to be a great deal of adaptation. Obviously if you're trying to target slightly different outcomes or trying to target a different population or working within a different delivery context, then I think that's when you need to step back and say are [more extensive] adaptations necessary and how do we go about finding out about that."*

*P013 researcher micro-in progress*

There was an expressed need for more comprehensive and systematic guidance to make assessments, with some individuals being particularly keen on recommendations on how to conceptualise and measure fidelity, and how to judge if intervention adaptation had transitioned into de novo development work:

*"If there's some kind of, some kind of scale, sort of like a fidelity scale, not fidelity obviously, but like a scale, then each time you had a new component, that's a mark off, and if there's a*

*specific cut off, where you know, the interventions significantly changed from when it was adapted, that'd be kind of complex to develop that. Something like that yeah."*

*P019 researcher meso-in progress*

## 4.0 Discussion

As research on the adaptation, implementation, and re-evaluation of interventions in new contexts rapidly progresses [1, 3, 4, 36], there is a need for guidance to support decision-making and conduct. Data from the qualitative study, which are presented in this paper, are imperative in ensuring that the guidance is sensitive and responsive to the complexity, challenges and uncertainties associated with the real-world practice of undertaking adaptation. One of the central findings to emerge from the data is a lack of shared understanding and clarity about the very meaning of adaptation. There was clear conceptual overlap with related constructs, such as implementation, and translation. This resonates with the findings from the ADAPT study's systematic review of adaptation frameworks, which found it being used interchangeably with concepts such as reinvention and transcreation [2].

Intersecting with this confusion was ambiguity about how intervention adaptation is different from the phases of intervention description and development [20] or from post-evaluation implementation, and its expansive literature within implementation science [37, 38]. Recently there has been emergence of the terms scale-out and scale-up to support the differentiation between adaptation and implementation research [39, 40]. The former relates to the transfer of interventions to contexts, while the latter refers to the expansion of implementation within the same context. While ostensibly helpful, there remains challenges with such a distinction. For example, recent research has highlighted that scale-up usually involves the intervention being delivered in a new temporal or spatial context that is not identical to the trial sample, who participated at an earlier point in the history of the system [39].

Through this exploration of definitions, it is evident that those working in the field feel it would be beneficial to have some coherence in key concepts, and parameters around types of evaluation research, in order to navigate the extant literature with more ease. Some standardisation may support the wider uptake of findings across diverse stakeholders. It may also increase the ability to evaluate adaptations. For example, a recent review, found that the impact of adaptations could not be assessed because reporting of the process was so unclear [12]. The recent MADI guidance offering direction here to enhance reporting [3], allowing for robust evaluation and for systematic synthesis within reviews.

Primarily participants considered the differences between contexts and the possible non-transferability of the evidence base as the rationale for adapting an intervention. This foregrounding of context seemingly reflects the growing prominence of realist evaluation and complex systems thinking within evaluation research [41–44] with both recognising the contextual contingency of effects. Yet despite widespread agreement that responding to the needs of the new context is a priority, there was a lack of certainty about how to actually assess contextual differences. There have been a range of frameworks issued in recent years to map contextual characteristics [8, 9], in addition to theoretical work from rugged landscape theory to help understand the uniqueness of system dynamics [45], and models to assess the need for re-evaluation in new contexts [40]. However, in practice these are not yet being used in a systematic manner. Even where they start to gain traction, it is important to recognise the risk of adaptation hyperactivity, where there is a focus on privileging contextual dissimilarities rather than similarities, perhaps in the effort to assert the presence of social equipoise [46]. Equally it will be challenging to prioritise the aspects of contexts to adapt to, as different stakeholders

will likely have different understandings of the problem and its relationship to the wider system [44]. Overall it was reflected by researchers that adaptation can be resource and time intensive similar to that of creating de novo interventions. However, funders felt that adaptation is more efficient and less resource intensive than de novo development. This is similar to the Delphi study findings where participants were divided about whether adaptation is quicker than de novo development and is an important motivator for adapting existing interventions [47]. This was one of the main areas of disagreement.

There was further consideration around the timing of adaptation. The literature tends to suggest that adaptations should be planned, often in advance of the intervention being implemented in a new context. However, study participants recognised the value of both planned and responsive adaptation, as it may only be when the intervention is interacting with the new systems' features that it is evident what modifications are necessary [2]. This chimes with more recent research that has recognised the role of reactive, responsive adaptation [3, 36, 48]. However, it is recognised the spontaneous adaptations that are not accordance the theory of change, can be limited where there has been careful, pre-planned adaptation to achieve intervention-context fit [49]. Valuing responsive adaptations may also be important in achieving a more bottom-up approach, with local stakeholders in the new context having a more central role in the process [50]. Again, in the wider literature there have been some advances in how to engage stakeholders in adaptations. For example, Analytic Hierarchy Process (AHP) has been developed to support the engagement of the target population in cultural adaptation [51]. However, such processes were not cited in the interview data, either in terms of engaging stakeholders in planned or responsive adaptation.

The study also provides important insight into stakeholders' views on fidelity in relation to adapted interventions. Within intervention evaluation research there has been a tendency to construct a binary position. From one perspective, there has been a focus on adherence to the form of interventions, or core components. The ADAPT studies systematic review of frameworks indicated that this is the predominant perspective used in adaptation research to date [2]. Alternatively, and particularly amongst those working from a more complex systems perspective, there is a focus on retaining the function or integrity of the intervention [17]. Here all components are eligible for adaptation provided that the intervention still activates the same causal mechanisms. The qualitative data indicated that this binary often collapsed in practice. In principle, participants maintained that it was the mechanisms of change that constituted the essence of the intervention and should not be modified, suggesting a focus on retaining form. This focus on mechanisms of change might also help clarify the difference between adaptation and de novo development, with changes to the underpinning mechanisms marking a transition to the development of a new intervention. However, it seemed difficult to operationalise a somewhat intangible construct in practice, and so many relied on the notion of core components to explain how the mechanisms could be retained in reality. This suggests the need for more consideration of how functional fidelity is operationalised. Although some guidance and worked examples are emerging, these are not yet in common use [52–54]. It may also be helpful to have improved reporting of both intervention mechanisms and components to support stakeholders in making fully informed decisions about what can be adapted [55, 56].

As the qualitative study has informed development of the ADAPT guidance [47], there are important considerations around the continued role of qualitative research as the guidance is disseminated and used in practice. There was a clearly expressed need for guidance from study participants, and the Delphi consensus exercise demonstrated a high degree of consensus across a range of items, indicating that some shared understandings and agreed processes could be identified [47]. However, while this is constructive in giving the field of adaptation

research an initial sense of coherency in the short-term, this sense of unified thinking should not be overestimated. In drawing out fractures, uncertainty and a lack of clarity in parts, the qualitative data has shown the extent of complexity that surrounds the guidance. This has been demonstrated through the disagreement across the stakeholders about the definition of and the rational for adaptation. In a field of research that is quickly evolving, and operates across a range of disciplines (e.g. health services, global health), it is important to engage with the nuance and diversity in perspectives on an ongoing basis as they will also likely change. In particular, future research might focus on exploring the use guidance in real-world contexts and how it has shaped understandings and experiences of adaptation, which may inform its future progress.

### 4.1 Study limitations

The study was primarily limited by the diversity of views. First, despite efforts to recruit PPI and policymakers, there was no participation from these stakeholder groups. As such their perspective, which may contrast with the generated data, were not included. The lack of policymakers also means that there is minimal insight into how intervention adaptation is commissioned and resourced at a national and local level. Second, while aiming to sample a diverse range of interventions (i.e. micro, meso, and macro), based on the assumption that they may interact with contexts differently and hence warrant different adaptations, there was a dearth of macro-level interventions [57]. This may be a consequence of such interventions, notably national policies, not being explicitly framed as adaptations even when derived from the principles and practices of interventions that are implemented elsewhere. Regardless of these limitations, the data did capture a varied and nuanced range of perspectives in relation to intervention adaptation.

### 4.2 Conclusions

Guidance to support the adaptation, implementation and re-evaluation of interventions in new contexts is important. In exploring the understandings of stakeholders, this study has highlighted the lack of clarity around key concepts and uncertainty about central decision-making processes, notably why interventions should be adapted, when and to what extent. This analysis informs the guidance developed as part of the ADAPT study [47]. This work can be applied to all adaptation areas. It is relevant to the current adaptation of interventions during COVID and to public health and global health in countries with a limited budget where a focused adaptation is required for their context. Future research needs to monitor how perspectives and understandings evolve, particularly as the guidance is used in real world practice. This will help to support future development and iterations of the guidance, to ensure it is responsive to the everchanging context of evaluation research.

## Supporting information

**S1 Appendix. Interview schedule: Researchers.**
(DOCX)

## Acknowledgments

We acknowledge the members of the wider ADAPT study team: Dr Mhairi Campbell, Prof Peter Craig, Dr Ani Movsisyan, Prof Alicia O'Cathain, Dr Lisa Pfadenhauer, Prof Eva Rehfuess, Dr Britt Hallingberg, Dr Laura Arnold, and Prof Laurence Moore. We also acknowledge the contributions of our research participants.

## Author Contributions

**Conceptualization:** Hannah Littlecott, Pat Hoddinott, Jeremy Segrott, Simon Murphy, Graham Moore, Rhiannon Evans.

**Data curation:** Lauren Copeland, Danielle Couturiaux.

**Formal analysis:** Lauren Copeland, Hannah Littlecott, Danielle Couturiaux, Rhiannon Evans.

**Funding acquisition:** Hannah Littlecott, Pat Hoddinott, Jeremy Segrott, Simon Murphy, Graham Moore, Rhiannon Evans.

**Methodology:** Rhiannon Evans.

**Project administration:** Lauren Copeland, Hannah Littlecott, Rhiannon Evans.

**Supervision:** Rhiannon Evans.

**Visualization:** Lauren Copeland, Rhiannon Evans.

**Writing – original draft:** Lauren Copeland, Rhiannon Evans.

**Writing – review & editing:** Lauren Copeland, Hannah Littlecott, Danielle Couturiaux, Pat Hoddinott, Jeremy Segrott, Simon Murphy, Graham Moore, Rhiannon Evans.

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
