## [Decision Letter · Decision Letter 0]

5 May 2021

PONE-D-20-38736

The what, why and when of adapting interventions for new contexts: A qualitative study of researchers, funders, journal editors and practitioners’ understandings

PLOS ONE

Dear Dr. Copeland,

Thank you for submitting your manuscript to PLOS ONE. After careful consideration, we feel that it has merit but does not fully meet PLOS ONE’s publication criteria as it currently stands. Therefore, we invite you to submit a revised version of the manuscript that addresses the points raised during the review process.

We look forward to receiving your revised manuscript.

Kind regards,

Ruth Jepson, PhD

Academic Editor

PLOS ONE

Journal Requirements:

Reviewers' comments:

Reviewer's Responses to Questions

**Comments to the Author**

1. Is the manuscript technically sound, and do the data support the conclusions?

Reviewer #1: Yes

Reviewer #2: Yes

2. Has the statistical analysis been performed appropriately and rigorously? 

Reviewer #1: N/A

Reviewer #2: Yes

3. Have the authors made all data underlying the findings in their manuscript fully available?

Reviewer #1: Yes

Reviewer #2: Yes

4. Is the manuscript presented in an intelligible fashion and written in standard English?

Reviewer #1: Yes

Reviewer #2: Yes

5. Review Comments to the Author

Reviewer #1: Thank you for the opportunity to review this paper. Having recent experience of adaptation, I appreciate the complexity and need to move forward this area of work. The findings and wider program of work will be of great interest to those looking to adapt public health interventions for new contexts.

I found the manuscript to be very well written and considered. I have only minor comments for the authors clarification/consideration:

• It is unclear what data analysis (line 109) is being referred to

• The authors note that participants were sampled until ‘saturation’ was reached, defining this as depth and representation across the spectrum of stakeholders, yet also report that no PPI or policymakers were recruited. Does this suggest that saturation was not reached?

• I was surprised that so few practitioners and policymakers participated in the interviews and wondered if this is reflective of the nature of the sampled case studies and how practitioners were involved in the adaptation process. Did the authors consider adaptations not published in the peer-reviewed literature and trial registries?

• On line 171 where data saturation is discussed I was unsure why this comes before recruitment of other stakeholders

• Could the authors provide a brief summary of what key findings and evidence gaps from the systematic and scoping reviews were addressed in the topic guides?

• Did the participants also participate in the Delphi study? (line 214-215 seems to suggest this).

• I was interested that participants commented on adaptation being more efficient than intervention development yet didn’t discuss the time and resources required to adapt, which can also be resource intensive.

• The authors note that a 10% sample of the data was independently checked by a second researcher. Were any errors identified or amendments made following this?

• Can the authors provide a justification for using the Framework approach?

• The findings were informative and well-written. Were there any differences in perspectives between the various stakeholders?

• Typos/referencing errors: lines 217, 247, 363, 455, 535, 536, 585, 612

Reviewer #2: This is a very well written paper and covers an important and topical issue in public health globally. The authors are to be commended on their qualitative methodology which in this case has produced rich insights to guide future practice. Given the timing of this paper, the authors could also highlight the importance of this work in ensuring non communicable diseases and other risk factors are also given due focus and attention and not lost in the sole focus on Covid. Studies such as this could help countries and organisations focus their limited budgets on the breadth of public health issues by ensuring evidence from other interventions is duly translated into locally relevant practice not only to meet local needs but also reduce duplication of spend and efforts in already stretched health and social disciplines. Congratulations on this important work.

6. PLOS authors have the option to publish the peer review history of their article (what does this mean?). If published, this will include your full peer review and any attached files.

Reviewer #1: No

Reviewer #2: **Yes: **Louise Baldwin

---

## [Author Response · Author response to Decision Letter 0]

10 Jun 2021

Please see uploaded covering letter which includes all responses to the reviews comments. We thank the reviewers for their comments which have helped to strengthen the paper.

---

## [Editor Report · Decision Letter 1]

18 Jun 2021

The what, why and when of adapting interventions for new contexts: A qualitative study of researchers, funders, journal editors and practitioners’ understandings

PONE-D-20-38736R1

Dear Dr. Copeland,

We’re pleased to inform you that your manuscript has been judged scientifically suitable for publication and will be formally accepted for publication once it meets all outstanding technical requirements.

Kind regards,

Ruth Jepson, PhD

Academic Editor

PLOS ONE

---

## [Editor Report · Acceptance letter]

1 Jul 2021

PONE-D-20-38736R1 

The what, why and when of adapting interventions for new contexts: A qualitative study of researchers, funders, journal editors and practitioners’ understandings 

Dear Dr. Copeland:

I'm pleased to inform you that your manuscript has been deemed suitable for publication in PLOS ONE. Congratulations! Your manuscript is now with our production department. 

Kind regards, 

on behalf of

Dr. Ruth Jepson 

Academic Editor

PLOS ONE